# 3D-Aware Hypothesis & Verification for Generalizable Relative Object Pose Estimation

**Chen Zhao**
EPFL-CVLab
chen.zhao@epfl.ch

**Tong Zhang** *
EPFL-IVRL
tong.zhang@epfl.ch

**Mathieu Salzmann**
EPFL-CVLab, ClearSpace SA
mathieu.salzmann@epfl.ch

## Abstract

Prior methods that tackle the problem of generalizable object pose estimation highly rely on having dense views of the unseen object. By contrast, we address the scenario where only a single reference view of the object is available. Our goal then is to estimate the relative object pose between this reference view and a query image that depicts the object in a different pose. In this scenario, robust generalization is imperative due to the presence of unseen objects during testing and the large-scale object pose variation between the reference and the query. To this end, we present a new hypothesis-and-verification framework, in which we generate and evaluate multiple pose hypotheses, ultimately selecting the most reliable one as the relative object pose. To measure reliability, we introduce a 3D-aware verification that explicitly applies 3D transformations to the 3D object representations learned from the two input images. Our comprehensive experiments on the Objaverse, LINEMOD, and CO3D datasets evidence the superior accuracy of our approach in relative pose estimation and its robustness in large-scale pose variations, when dealing with unseen objects. Our project website is at: https://sailor-z.github.io/projects/ICLR2024_3DAHV.html.

## 1 Introduction

Object pose estimation is crucial in many computer vision and robotics tasks, such as VR/AR (Azuma, 1997), scene understanding (Geiger et al., 2012; Chen et al., 2017; Xu et al., 2018; Marchand et al., 2015), and robotic manipulation (Collet et al., 2011; Zhu et al., 2014; Tremblay et al., 2018; Pitteri et al., 2019). Much effort has been made toward estimating object pose parameters either by direct regression (Xiang et al., 2017; Wang et al., 2019a; Hu et al., 2020) or by establishing correspondences (Peng et al., 2019; Wang et al., 2021; Su et al., 2022) which act as input to a PnP algorithm (Lepetit et al., 2009). These methods have achieved promising results in the closed-set scenario, where the training and testing data contain the same object instances. However, this assumption restricts their applicability to the real world, where unseen objects from new categories often exist. Therefore, there has been growing interest in generalizable object pose estimation, aiming to develop models that generalize to unseen objects in the testing phase.

In this context, some approaches (Zhao et al., 2022b; Shugurov et al., 2022) follow a template-matching strategy, matching a query object image with reference images generated by rendering the 3D textured object mesh from various viewpoints. To address the scenario where the object mesh is unavailable, as illustrated in Fig. 1(a), some methods take real dense-view images as references. The object pose in the query image is estimated either by utilizing a template-matching mechanism (Liu et al., 2022) or by building 2D-3D correspondences (Sun et al., 2022). A computationally expensive 3D reconstruction (Schonberger & Frahm, 2016) is involved to either calibrate the reference images or reconstruct the 3D object point cloud. In any event, the requirement of dense-view references precludes the use of these methods for individual or sparse images, e.g., downloaded from the Internet. Intuitively, with sufficiently diverse training data, one could think of learning to regress the object pose parameters directly from a single query image. However, without access to a canonical object frame, the predicted object pose would be ill-defined as it represents the relative transformation between the camera frame and the object frame.

---

*Corresponding author.

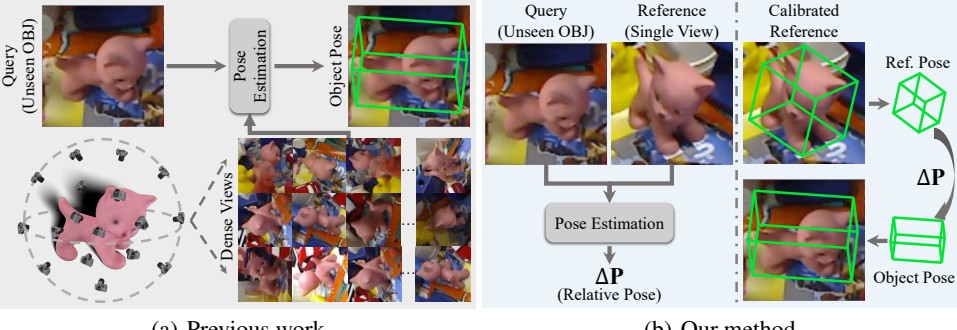

(a) Previous work                (b) Our method

Figure 1: **Difference between previous work and our method.** Previous approaches (a) estimate the pose of an unseen object building upon either template matching or 2D-3D correspondences, which requires dense views of the object as references. By contrast, our method (b) takes only one reference as input and predicts the relative object pose between the reference and query. The object pose in the query can be derived when the pose of the reference is available.

To bypass this issue, we assume the availability of a single reference image that contains the novel object. As shown in Fig. 1(b), we take this reference to be the canonical view and estimate the relative object pose between the reference view and the query view, which is thus well-defined. If the object pose in the reference is provided, the object pose in the query can be derived. In this scenario, one plausible solution is to compute the relative object pose based on pixel-level correspondences (Lowe, 2004; Rublee et al., 2011). However, the two views may depict a large-scale object pose variation, and our experiments will evidence that even the state-of-the-art feature-matching approaches (Sarlin et al., 2020; Sun et al., 2021; Goodwin et al., 2022) cannot generate reliable correspondences in this case, which thus results in inaccurate relative object pose estimates. As an alternative, Zhang et al. (2022); Lin et al. (2023) predict the likelihood of pose parameters leveraging an energy-based model, which, however, lacks the ability to capture 3D information when learning 2D feature embeddings.

By contrast, we adopt a *hypothesis-and-verification* paradigm, drawing inspiration from its remarkable success in robust estimation (Fischler & Bolles, 1981). We randomly sample pose parameter hypotheses and verify the reliability of these hypotheses. The relative object pose is determined as the most reliable hypothesis. Since relative pose denotes a 3D transformation, achieving robust verification from two 2D images is non-trivial. Our innovation lies in a 3D-aware verification mechanism. Specifically, we develop a 3D reasoning module over 2D feature maps, which infers 3D structural features represented as 3D volumes. This lets us explicitly apply the pose hypothesis as a 3D transformation to the reference volume. Intuitively, the transformed reference volume should be aligned with the query one if the sampled hypothesis is correct. We thus propose to verify the hypothesis by comparing the feature similarities of the reference and the query. To boost robustness, we aggregate the 3D features into orthogonal 2D plane embeddings and compare these embeddings to obtain a similarity score that indicates the reliability of the hypothesis.

Our method achieves state-of-the-art performance on an existing benchmark of Lin et al. (2023). Moreover, we extend the experiments to a new benchmark for **g**eneralizable **r**elative **o**bject **p**ose estimation, which we refer to as GROP. Our benchmark contains over 10,000 testing image pairs, exploiting objects from Objaverse (Deitke et al., 2023) and LINEMOD (Hinterstoisser et al., 2012) datasets, thus encompassing both synthetic and real images with diverse object poses. In the context of previously unseen objects, our method outperforms the feature-matching and energy-based techniques by a large margin in terms of both relative object pose estimation accuracy and robustness. We summarize our contributions as follows:

- We highlight the importance of relative pose estimation for novel objects in scenarios where only one reference image is available for each object.

- We present a new hypothesis-and-verification paradigm where verification is made aware of 3D by acting on a learnable 3D object representation.

- We develop a new benchmark called GROP, where the evaluation of relative object pose estimation is conducted on both synthetic and real images with diverse object poses.

## 2 RELATED WORK

**Instance-Specific Object Pose Estimation.** The advancements in deep learning have revolutionized the field of object pose estimation. Most existing studies have focused on instance-level object pose estimation (Xiang et al., 2017; Peng et al., 2019; Wang et al., 2021; Su et al., 2022; Wang et al., 2019a), aiming to determine the pose of specific object instances. These methods have achieved remarkable performance in the closed-set setting, which means that the training data and testing data contain the same object instances. However, such an instance-level assumption restricts the applications in the real world where previously unseen objects widely exist. The studies of Zhao et al. (2022b); Liu et al. (2022) have revealed the limited generalization ability of the instance-level approaches when confronted with unseen objects. Some approaches (Wang et al., 2019b; Chen et al., 2020a; Lin et al., 2022) relaxed the instance-level constraint and introduced category-level object pose estimation. More concretely, the testing and training datasets consist of different object instances but the same object categories. As different instances belonging to the same category depict similar visual patterns, the category-level object pose estimation methods are capable of generalizing well to new instances. However, these approaches still face challenges in generalizing to objects from novel categories, since the object appearance could vary significantly.

**Generalizable Object Pose Estimation.** Recently, some effort has been made toward generalizable object pose estimation. The testing data may include objects from categories that have not been encountered during training. The objective is to estimate the pose of these unseen objects without retraining the network. In such a context, the existing approaches can be categorized into two groups, i.e., template-matching methods (Sundermeyer et al., 2020; Labbé et al., 2022; Zhao et al., 2022b; Liu et al., 2022; Shugurov et al., 2022) and feature-matching methods (Sun et al., 2022; He et al., 2022b). Given a query image of the object, the template-matching methods retrieve the most similar reference image from a pre-generated database. The object pose is taken as that in the retrieved reference. The database is created by either rendering the 3D object model or capturing images from various viewpoints. The feature-matching methods reconstruct the 3D object point cloud by performing SFM (Schonberger & Frahm, 2016) over a sequence of images. The 2D-3D matches are then built over the query image and the reconstructed point cloud, from which the object pose is estimated by using the PnP algorithm. Notably, these two groups both require dense-view reference images to be available. Therefore, they cannot be applied in scenarios where only sparse images are accessible.

**Relative Object Pose Estimation.** Some existing methods could nonetheless be applied for relative object pose estimation, even though they were designed for a different purpose. For example, one could use traditional (Lowe, 2004) or learning-based (Sarlin et al., 2020; Sun et al., 2021; Goodwin et al., 2022) methods to build pixel-pixel correspondences and compute the relative pose by using multi-view geometry (Hartley & Zisserman, 2003). However, as only two views (one query and one reference) are available, large-scale object pose variations are inevitable, posing challenges to the correspondence-based approaches. Moreover, RelPose (Zhang et al., 2022) and RelPose++ (Lin et al., 2023) build upon an energy-based model, which combines the pose parameters with the two-view images as the input and predicts the likelihood of the relative camera pose. However, RelPose and RelPose++ exhibit a limitation in their ability to reason about 3D information, which we found crucial for inferring the 3D transformation between 2D images. By contrast, we propose to explicitly utilize 3D information in a new hypothesis-and-verification paradigm, achieving considerably better performance in our experiments.

## 3 METHOD

### 3.1 PROBLEM FORMULATION

We train a network on *RGB* images depicting specific object instances from a set $\mathcal{O}_{train}$. During testing, we aim for the network to generalize to new objects in the set $\mathcal{O}_{test}$, with $\mathcal{O}_{test} \cap \mathcal{O}_{train} = \emptyset$. In contrast to some previous methods which assume that $\mathcal{O}_{train}$ and $\mathcal{O}_{test}$ contain the same cate-

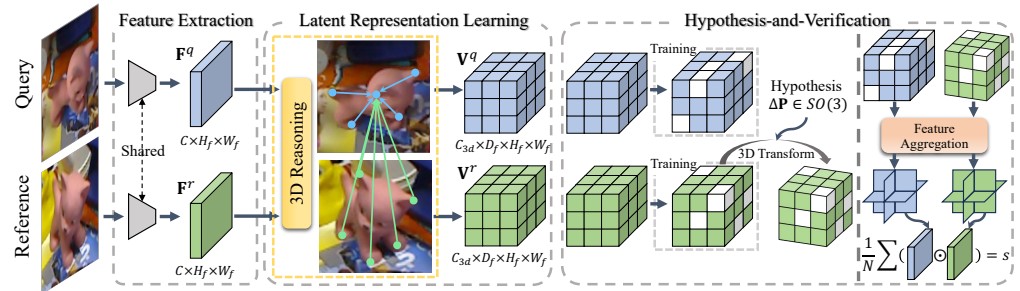

Figure 2: **Overview of our framework.** Our method estimates the relative pose of previously unseen objects given two images, building upon a new hypothesis-and-verification paradigm. A hypothesis $\Delta \mathbf{P}$ is randomly sampled and its accuracy is measured as a score $s$. To explicitly integrate 3D information, we perform the verification over a 3D object representation indicated as a learnable 3D volume. The sampled hypothesis is coupled with the learned representation via a 3D transformation over the reference 3D volume. We learn the 3D volumes from the 2D feature maps extracted from the RGB images by introducing a 3D reasoning module. To improve robustness, we randomly mask out some blocks colored in white during training.

gories, i.e., $\mathcal{C}_{train} = \mathcal{C}_{test}$, we work on generalizable object pose estimation. The testing objects in $\mathcal{O}_{test}$ may belong to previously unseen categories, i.e., $\mathcal{C}_{test} \neq \mathcal{C}_{train}$. In such a context, we propose to estimate the relative pose $\Delta \mathbf{P}$ of the object depicted in two images $\mathbf{I}_q$ and $\mathbf{I}_r$. As the 3D object translation can be derived by utilizing 2D detection (Saito et al., 2022; Wang et al., 2023; Kirillov et al., 2023), we focus on the estimation of 3D object rotation $\Delta \mathbf{R} \in SO(3)$, which is more challenging. As illustrated in Fig. 2, our method builds upon a hypothesis-and-verification mechanism (Fischler & Bolles, 1981). Concretely, we randomly sample an orientation hypothesis $\Delta \mathbf{R}_i$, utilizing the 6D continuous representation of Zhou et al. (2019). We then verify the correctness of $\Delta \mathbf{R}_i$ using a verification score $s_i = f(\mathbf{I}_q, \mathbf{I}_r | \Delta \mathbf{R}_i, \Theta)$, where $f$ indicates a network with learnable parameters $\Theta$. The expected $\Delta \mathbf{R}^*$ is determined as the hypothesis with the highest verification score, i.e.,

$$\Delta \mathbf{R}^* = \underset{\Delta \mathbf{R}_i \in SO(3)}{\arg\max} \; f(\mathbf{I}_q, \mathbf{I}_r | \Delta \mathbf{R}_i, \Theta) \; . \tag{1}$$

To facilitate the verification, we develop a 3D transforming layer over a learnable 3D object representation. The details will be introduced in this section.

### 3.2 3D OBJECT REPRESENTATION LEARNING

Predicting 3D transformations from 2D images is inherently challenging, as it necessitates the capability of 3D reasoning. Furthermore, the requirement of generalization ability to unseen objects makes the problem even harder. Existing methods (Zhang et al., 2022; Lin et al., 2023) tackle this challenge by deriving 3D information from global feature embeddings, which are obtained through global pooling over 2D feature maps. However, this design exhibits two key drawbacks: First, the low-level structural features which are crucial for reasoning about 3D transformations are lost; Second, the global pooling process incorporates high-level semantic information (Zhao et al., 2022b), which is coupled with the object category. Therefore, these approaches encounter difficulties in accurately estimating the relative pose of previously unseen objects.

To address this, we introduce a 3D object representation learning module that is capable of reasoning about 3D information from 2D structural features. Concretely, the process begins by feeding the query and reference images into a pretrained encoder (Ranftl et al., 2020), yielding two 2D feature maps $\mathbf{F}^q, \mathbf{F}^r \in \mathbb{R}^{C \times H_f \times W_f}$. As no global pooling layer is involved, $\mathbf{F}^q$ and $\mathbf{F}^r$ contain more structural information than the global feature embeddings of (Zhang et al., 2022; Lin et al., 2023). Subsequently, $\mathbf{F}^q$ and $\mathbf{F}^r$ serve as inputs to a 3D reasoning module. Since each RGB image depicts the object from a particular viewpoint, inferring 3D features from a single 2D feature map is intractable. To address this issue, we combine the query and reference views and utilize the transformer (Vaswani et al., 2017; Dosovitskiy et al., 2020), renowned for its ability to capture relationships among local patches.

Our 3D reasoning block comprises a self-attention layer and a cross-attention layer, which account for the intra-view and inter-view relationships, respectively. Notably, unlike the existing method of Lin et al. (2023) that utilizes transformers at an image level, i.e., treating a global feature embedding as a token, our module takes $\mathbf{F}^q$ and $\mathbf{F}^r$ as input, thereby preserving more structural information throughout the process. Specifically, we compute

$$\mathbf{F}^q_{l+1} = g(\mathbf{F}^q_l, \mathbf{F}^r_l | \Omega^q_{\text{self}}, \Omega^q_{\text{cross}}), \tag{2}$$

$$\mathbf{F}^r_{l+1} = g(\mathbf{F}^r_l, \mathbf{F}^q_l | \Omega^r_{\text{self}}, \Omega^r_{\text{cross}}), \tag{3}$$

where $g$ denotes the 3D reasoning block with learnable parameters $\{\Omega^q_{\text{self}}, \Omega^q_{\text{cross}}, \Omega^r_{\text{self}}, \Omega^r_{\text{cross}}\}$. Let us take $\mathbf{F}^q$ as an example as the process over $\mathbf{F}^r$ is symmetric. We tokenize $\mathbf{F}^q$ by flattening it from $\mathbb{R}^{C \times H_f \times W_f}$ to $\mathbb{R}^{N \times C}$, where $N = H_f \times W_f$. A position embedding (Dosovitskiy et al., 2020) is added to the sequence of tokens, which accounts for positional information. To ensure a broader receptive field that covers the entire object, the tokens are fed into a self-attention layer, which is formulated as $\tilde{\mathbf{F}}^q_l = t(\mathbf{F}^q_l, \mathbf{F}^q_l | \Omega^q_{\text{self}})$, where $t$ denotes the attention layer. As aforementioned, $\tilde{\mathbf{F}}^q_l$ only describes the object in $\mathbf{I}_q$, which is captured from a single viewpoint. We thus develop a cross-attention layer, incorporating information from the other view $\mathbf{I}_r$ into $\tilde{\mathbf{F}}^q_l$. We denote the cross attention as $\mathbf{F}^q_{l+1} = t(\tilde{\mathbf{F}}^q_l, \tilde{\mathbf{F}}^r_l | \Omega^r_{\text{cross}})$, where $\mathbf{F}^q_{l+1}$ serves as the input of the next 3D reasoning block.

We denote the output of the last 3D reasoning block as $\hat{\mathbf{F}}^q, \hat{\mathbf{F}}^r \in \mathbb{R}^{C \times H_f \times W_f}$. $\hat{\mathbf{F}}^q$ and $\hat{\mathbf{F}}^r$ comprise both intra-view and inter-view object-related information. Nevertheless, it is still non-trivial to couple the 3D transformation with the 2D feature maps, which is crucial in the following hypothesis-and-verification module. To handle this, we derive a 3D object representation from the 2D feature map in a simple yet effective manner. We lift $\hat{\mathbf{F}}^q$ and $\hat{\mathbf{F}}^r$ from 2D space to 3D space, i.e., $\mathbb{R}^{C \times H_f \times W_f} \rightarrow \mathbb{R}^{C_{3d} \times D_f \times H_f \times W_f}$, where $C = C_{3d} \times D_f$. The 3D representations are thus encoded as 3D volumes $\mathbf{V}^q$ and $\mathbf{V}^r$. Since the spatial dimensionality of $\mathbf{V}^q$ and $\mathbf{V}^r$ matches that of the 3D transformation, such a lifting process enables the subsequent 3D-aware verification.

## 3.3 3D-AWARE HYPOTHESIS AND VERIFICATION

The hypothesis-and-verification mechanism has achieved tremendous success as a robust estimator (Fischler & Bolles, 1981) for image matching (Yi et al., 2018; Zhao et al., 2021). The objective is to identify the most reliable hypothesis from multiple samplings. In such a context, an effective verification process is critical. Moreover, in the scenario of relative object pose estimation, we expect the verification to be differentiable and aware of the 3D transformation. We thus tailor the hypothesis-and-verification mechanism to meet these new requirements.

We develop a 3D masking approach in latent space before sampling hypotheses, drawing inspiration from the success of the masked visual modeling methods (He et al., 2022a; Xie et al., 2022). Instead of masking the RGB images, we propose to mask the learnable 3D volumes, which we empirically found more compact and effective. Specifically, we sample two binary masks $\mathbf{V}^q_b, \mathbf{V}^r_b \in \mathbb{R}^{C_{3d} \times D_f \times H_f \times W_f}$ during training, initialized as all ones. $h$ of the elements in each mask are randomly set to 0. 3D masking is performed as $\tilde{\mathbf{V}}^q = \mathbf{V}^q \odot \mathbf{V}^q_b, \tilde{\mathbf{V}}^r = \mathbf{V}^r \odot \mathbf{V}^r_b$, where $\odot$ denotes the Hadamard product. Note that the masking is asymmetric over $\mathbf{V}^q_b$ and $\mathbf{V}^r_b$. Such a design enables the modeling of object motion between two images (Gupta et al., 2023), offering potential benefits to the task of relative object pose estimation.

The hypothesis-and-verification process begins by randomly sampling hypotheses, utilizing the 6D continuous representation of Zhou et al. (2019). Each hypothesis is then converted to a 3D rotation matrix $\Delta \mathbf{R}_i$, i.e., $\mathbb{R}^6 \rightarrow \mathbb{R}^{3 \times 3}$. During the verification, we explicitly couple the hypothesis with the learnable 3D representation by performing a 3D transformation. This is formulated as

$$\tilde{\mathbf{V}}^r = \varphi(\Delta \mathbf{R}_i \mathbf{X}^r), \mathbf{X}^r \in \mathbb{R}^{3 \times L}, \tag{4}$$

where $\mathbf{X}^r$ denotes the 3D coordinates of the elements in $\tilde{\mathbf{V}}^r$ with $L = D_f \times H_f \times W_f$ and $\varphi$ indicates trilinear interpolation. We keep the query 3D volume unchanged and only transform the reference 3D volume. Intuitively, the transformed $\tilde{\mathbf{V}}^r$ should be aligned with $\tilde{\mathbf{V}}^q$ if the sampled hypothesis is correct. Conversely, an incorrect 3D transformation is supposed to result in a noticeable disparity between the two 3D volumes. Therefore, our transformation-based approach facilitates the verification of $\Delta \mathbf{R}_i$, which could be implemented by assessing the similarity between $\tilde{\mathbf{V}}^q$ and

$\tilde{\mathbf{V}}^r$. However, the transformed $\tilde{\mathbf{V}}^r$ tends to be noisy in practice because of zero padding during the transformation and some nuisances such as the background. We thus introduce a feature aggregation module, aiming to distill meaningful information for robust verification. More concretely, we project $\tilde{\mathbf{V}}^q$ and $\tilde{\mathbf{V}}^r$ back to three orthogonal 2D planes, i.e., $\mathbb{R}^{C_{3d} \times D_f \times H_f \times W_f} \rightarrow \mathbb{R}^{3C \times H_f \times W_f}$ with $C = C_{3d} \times D_f$, and aggregate the projected features as $\mathbf{A}^q = g(\tilde{\mathbf{V}}^q|\Psi), \mathbf{A}^r = g(\tilde{\mathbf{V}}^r|\Psi)$, where $\mathbf{A}^q, \mathbf{A}^r \in \mathbb{R}^{C_{2d} \times H_f \times W_f}$ represent the distilled feature embeddings and $g$ is the aggregation module with learnable parameters $\Psi$. The verification score is then computed as

$$s_i = \frac{1}{N} \sum_{j,k} \frac{\mathbf{A}^q_{jk} \cdot \mathbf{A}^r_{jk}}{\|\mathbf{A}^q_{jk}\| \cdot \|\mathbf{A}^r_{jk}\|}, \quad \mathbf{A}^q_{jk}, \mathbf{A}^r_{jk} \in \mathbb{R}^{C_{2d}}. \tag{5}$$

We run the hypothesis and verification $M$ times in parallel and the expected $\Delta\mathbf{R}^*$ is identified as

$$\Delta\mathbf{R}^* = \Delta\mathbf{R}_k, \quad k = \arg\max_i \{s_i, i = 1, 2, \ldots, M\}. \tag{6}$$

Note that compared with the dynamic rendering method (Park et al., 2020) which optimizes the object pose by rendering and comparing depth images, our approach performs verification in the latent space. This eliminates the need for computationally intensive rendering and operates independently of depth information. An alternative to the hypothesis-and-verification mechanism consists of optimizing $\Delta\mathbf{R}$ via gradient descent. However, our empirical observations indicate that this alternative often gets trapped in local optima. Moreover, compared with the energy-based approaches (Zhang et al., 2022; Lin et al., 2023), our method achieves a 3D-aware verification. To highlight this, let us formulate the energy-based model with some abuse of notation as

$$\Delta\mathbf{R}^* = \arg\max_{\Delta\mathbf{R}_i \in SO(3)} s_i, \quad s_i = \text{FC}(f(\mathbf{I}_q, \mathbf{I}_r) + h(\Delta\mathbf{R}_i)), \tag{7}$$

where FC denotes fully connected layers. In this context, the 2D image embedding and the pose embedding are learned as $f(\mathbf{I}_q, \mathbf{I}_r)$ and $h(\Delta\mathbf{R}_i)$, separately. By contrast, in our framework, the volume features are conditioned on $\Delta\mathbf{R}_i$ via the 3D transformation, which thus facilitates the 3D-aware verification.

We train our network using an infoNCE loss (Chen et al., 2020b), which is defined as

$$\mathcal{L} = -\log \frac{\sum_{j=1}^{P} \exp(s^p_j/\tau)}{\sum_{i=1}^{M} \exp(s_i/\tau)}, \tag{8}$$

where $s^p_j$ denotes the score of a positive hypothesis, and $\tau = 0.1$ is a predefined temperature. The positive samples are identified by computing the geodesic distance as

$$D = \arccos\left(\frac{\text{tr}(\Delta\mathbf{R}_i^{\mathsf{T}}\Delta\mathbf{R}_{\text{gt}}) - 1}{2}\right)/\pi, \tag{9}$$

where $\Delta\mathbf{R}_{\text{gt}}$ is the ground truth. We then consider hypotheses with $D < \lambda$ as positive samples.

## 4 EXPERIMENTS

### 4.1 IMPLEMENTATION DETAILS

In our experiments, we employ $4$ 3D reasoning blocks. We set the number of hypotheses during training and testing to $M = 9,000$ and $M = 50,000$, respectively. We define the masking threshold $h = 0.25$ and the geodesic distance threshold $\lambda = 15°$ (Zhang et al., 2022; Lin et al., 2023). We train our network for 25 epochs using the AdamW (Loshchilov & Hutter, 2017) optimizer with a batch size of $48$ and a learning rate of $10^{-4}$, which is divided by 10 after 20 epochs. Training takes around 4 days on 4 NVIDIA Tesla V100s.

### 4.2 EXPERIMENTAL SETUP

We compare our method with several relevant competitors including feature-matching methods, i.e., SuperGlue (Sarlin et al., 2020), LoFTR (Sun et al., 2021), and ZSP (Goodwin et al., 2022),

|  | SuperGlue | LoFTR | ZSP | Regress | RelPose | RelPose++ | **Ours** |
|---|---|---|---|---|---|---|---|
| Angular Error ↓ | 67.2 | 77.5 | 87.5 | 46.0 | 50.0 | 38.5 | **28.5** |
| Acc @ 30° (%) ↑ | 45.2 | 37.9 | 25.7 | 60.6 | 64.2 | 77.0 | **83.5** |
| Acc @ 15° (%) ↑ | 37.7 | 33.1 | 14.6 | 42.7 | 48.6 | 69.8 | **71.0** |

Table 1: **Experimental results on CO3D.**

|  | SuperGlue | LoFTR | ZSP | Regress | RelPose | RelPose++ | **Ours** |
|---|---|---|---|---|---|---|---|
| Angular Error ↓ | 102.4 | 134.1 | 107.2 | 55.9 | 80.4 | 33.5 | **28.1** |
| Acc @ 30° (%) ↑ | 15.1 | 9.6 | 4.2 | 39.2 | 20.8 | 72.3 | **78.6** |
| Acc @ 15° (%) ↑ | 12.1 | 7.7 | 1.5 | 15.6 | 6.7 | 42.9 | **58.4** |

Table 2: **Experimental results on Objaverse.**

|  | SuperGlue | LoFTR | ZSP | Regress | RelPose | RelPose++ | **Ours** |
|---|---|---|---|---|---|---|---|
| Angular Error ↓ | 64.8 | 84.5 | 78.6 | 52.1 | 58.3 | 46.6 | **41.7** |
| Acc @ 30° (%) ↑ | 26.2 | 24.2 | 10.7 | 26.5 | 26.1 | 42.5 | **61.5** |
| Acc @ 15° (%) ↑ | 14.3 | 13.5 | 2.7 | 7.6 | 7.0 | 15.8 | **29.9** |

Table 3: **Experimental results on LINEMOD.**

energy-based methods, i.e., RelPose (Zhang et al., 2022) and RelPose++ (Lin et al., 2023), and a regression method (Lin et al., 2023). We first perform an evaluation using the benchmark defined in (Lin et al., 2023), where the experiments are conducted on the CO3D (Reizenstein et al., 2021) dataset. We report the angular error between the predicted $\Delta\mathbf{R}$ and the ground truth, which is computed as in Eq. 9, and the accuracy with thresholds of 30° and 15° (Zhang et al., 2022; Lin et al., 2023). Furthermore, We extend the evaluation by introducing a new benchmark called GROP. To this end, we utilize the Objaverse (Deitke et al., 2023) and LINEMOD (Hinterstoisser et al., 2012) datasets, which include synthetic and real data, respectively. We retrain RelPose, RelPose++, and the regression method in our benchmark, and use the pretrained models for SuperGlue and LoFTR since retraining these two feature-matching approaches requires additional pixel-level annotations. For ZSP, as there is no training process involved, we evaluate it using the code released by the authors. We derive $\Delta\mathbf{R}$ from the estimated essential matrix (Hartley & Zisserman, 2003) for the feature-matching methods because we only have access to RGB images. We evaluate all methods on identical predefined query and reference pairs (8,304 on Objaverse and 5,000 on LINEMOD), which ensures a fair comparison. Given our emphasis on relative object rotation estimation, we crop the objects from the original RGB image utilizing the ground-truth object bounding boxes (Xiao et al., 2019; Zhao et al., 2022b; Park et al., 2020; Nguyen et al., 2022). In Sec. 4.5, we evaluate robustness against noise in the bounding boxes.

### 4.3 EXPERIMENTS ON CO3D

Let us first evaluate our approach in the benchmark used in (Zhang et al., 2022; Lin et al., 2023), which builds upon the CO3D dataset (Reizenstein et al., 2021). All testing objects here are previously unseen and the evaluation thus emphasizes the generalization ability. Table 1 reports the results in terms of angular error and accuracy. Note that the results of SuperGlue, Regress, RelPose, and RelPose++ shown here align closely with the ones reported in (Lin et al., 2023), lending credibility to the evaluation. More importantly, our method produces consistently more precise relative object poses, with improvements of at least **10%** in angular error. This evidences the generalization ability of our approach to unseen objects.

### 4.4 EXPERIMENTS ON GROP

Let us now develop the evaluation in our benchmark. Table 2 and Table 3 provide the experimental results on Objaverse and LINEMOD, respectively. Our method also achieves superior generalization ability to unseen objects, outperforming the previous methods by a substantial margin. For instance,

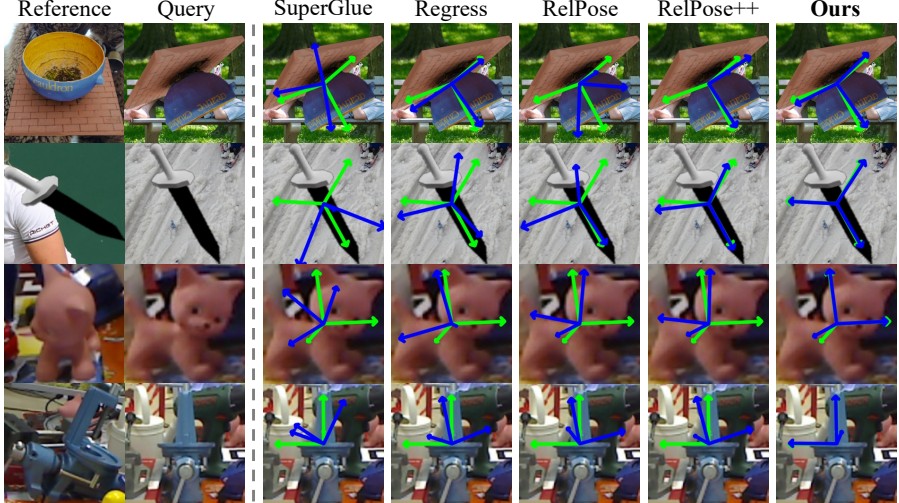

Figure 3: **Qualitative results on Objaverse and LINEMOD.** Here, we assume the reference to be calibrated and visualize the object pose in the query, which is derived from the estimated relative object pose. The predicted and ground-truth object poses are indicated by blue and green arrows, respectively.

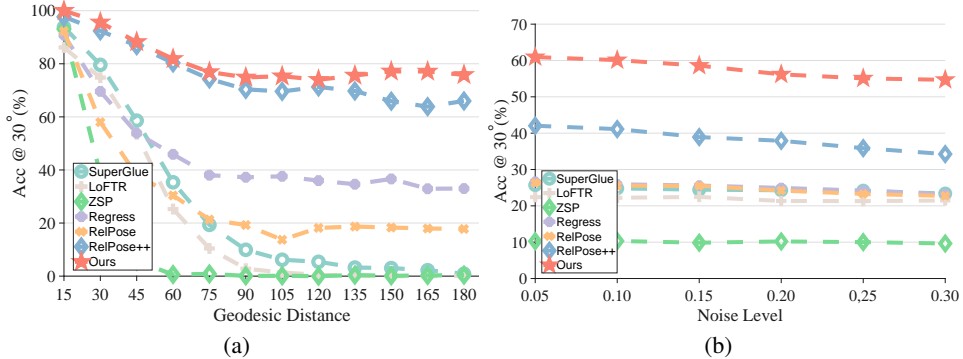

Figure 4: **Robustness.** (a) Acc @ $30°$ curves obtained with varying degrees of object pose variation between the reference and the query, measured by the geodesic distance. (b) Similar curves but for different levels of noise added to the object bounding boxes.

we achieve an improvement of at least **15.5%** on Objaverse and **14.1%** on LINEMOD, measured in terms of Acc @ $15°$. Moreover, we illustrate some qualitative results in Fig. 3. To this end, we assume the object pose $\mathbf{R}^r$ in the reference to be available, and the object pose $\mathbf{R}^q$ in the query is computed as $\mathbf{R}^q = \Delta\mathbf{R}\mathbf{R}^r$. We represent the predicted and the ground-truth object poses as blue and green arrows, respectively. This evidences that our method consistently yields better predictions. In the scenario where there is a notable difference in object pose between the reference and query (as in the cat images in the third row), the previous methods struggle to accurately predict the pose for the unseen object, while our approach continues to deliver an accurate prediction.

## 4.5 ABLATION STUDIES

To shed more light on the superiority of our method, we develop comprehensive ablation studies on Objaverse and LINEMOD. Most of the experiments are conducted on LINEMOD since it is a real dataset. As the two sparse views, i.e., a reference and a query, might result in a large-scale object pose variation, we start the ablations by analyzing the robustness in such a context. Specifically, we divide the Objaverse testing data into several groups based on the object pose variation between the reference and query, measured by geodesic distance. The task becomes progressively more

|  | w/o att. | w/o mask | w/ 2D mask | w/o agg. | RelPose* | **Ours** |
|---|---|---|---|---|---|---|
| Angular Error $\downarrow$ | 41.9 | 42.1 | 42.6 | 41.9 | 59.7 | **41.7** |
| Acc @ $30°$ (%) $\uparrow$ | 60.0 | 59.6 | 60.1 | 59.4 | 26.4 | **61.5** |
| Acc @ $15°$ (%) $\uparrow$ | 28.2 | 27.9 | 27.3 | 26.4 | 7.9 | **29.9** |

Table 4: **Effectiveness of the key components in our pipeline.**

challenging as the distance increases. We developed this experiment on Objaverse because of its wider range of pose variations compared to LINEMOD. Fig. 4(b) shows the Acc @ $30°$ curves as the distance varies from $0°$ to $180°$. Note that all methods demonstrate satisfactory predictions when the distance is small, i.e., when the object orientations in the reference and query views are similar. However, the performance of feature-matching approaches, i.e., SuperGlue, LoFTR, and ZSP, dramatically drops as the distance increases. This observation supports our argument that the feature-matching methods are sensitive to the pose variations. By contrast, our method consistently surpasses all competitors, thus showing better robustness.

As the object bounding boxes obtained in practice are inevitably noisy, we evaluate the robustness against the noise in this context on LINEMOD. Concretely, we add noise to the ground-truth bounding boxes by applying jittering to both the object center and scale. The jittering magnitude varies from $0.05$ to $0.30$, which results in different levels of noise. The experimental results are shown in Fig. 4(b), where our method outperforms the competitors across all scenarios. This promising robustness underscores the possibility of integrating our method with existing unseen object detectors (Zhao et al., 2022a; Liu et al., 2022). To showcase this, we extend our method to 6D unseen object pose estimation by combining it with the detector introduced in (Liu et al., 2022) and provide some results in the appendix.

Furthermore, we evaluate the effectiveness of the key components in our framework. The results on LINEMOD are summarized in Table 4, where the evaluation of effectiveness encompasses four distinct aspects: First, we develop a counterpart by excluding self-attention and cross-attention layers (w/o att.) from the 3D reasoning blocks; Second, we modify the 3D masking by either omitting it (w/o mask) or substituting it with a 2D masking process over RGB images (w/ 2D mask); Third, we directly compute the similarity of 3D volumes without utilizing the 2D aggregation module (w/o agg.); Fourth, we replace our 3D-aware verification mechanism with the energy-based model (Zhang et al., 2022; Lin et al., 2023) (RelPose*), while retaining our feature extraction backbone unchanged. The modified versions, namely w/o att., w/o mask, w/ 2D mask, and w/o agg., exhibit worse performance, which thus demonstrates the effectiveness of the presented components, i.e., attention layers, 3D masking, and the feature aggregation module. Additionally, the inferior results yield by RelPose* highlight that the 3D-aware verification mechanism contributes to the high-accuracy predictions, instead of the feature extraction backbone in our framework. Consequently, this observation supports our claim that the proposed verification module facilitates the relative pose estimation for unseen objects by preserving the structural features and explicitly utilizing 3D information.

## 5 CONCLUSION

In this paper, we have tackled the problem of relative pose estimation for unseen objects. We assume the availability of only one object image as the reference and aim to estimate the relative object pose between the reference and a query image. In this context, we have tailored the hypothesis-and-verification paradigm by introducing a 3D-aware verification, where the 3D transformation is explicitly coupled with a learnable 3D object representation. We have developed comprehensive experiments on Objaverse, LINEMOD, and CO3D datasets, taking both synthetic and real data with diverse object poses into account. Our method remarkably outperforms the competitors across all scenarios and achieves better robustness against different levels of object pose variations and noise. Since our verification module incorporates local similarities when computing the verification scores, it could be affected by the occlusions. This stands as a potential limitation that we consider, and we intend to explore and address this issue in our future research endeavors.

## ACKNOWLEDGMENT

This work was funded in part by the Swiss National Science Foundation via the Sinergia grant CRSII5-180359 and the Swiss Innovation Agency (Innosuisse) via the BRIDGE Discovery grant 40B2-0_194729.

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

APPENDIX

## A  ARCHITECTURE OF THE 3D REASONING MODULE

We show the architecture of the 3D reasoning module in Fig. 5. Each 3D reasoning block consists of a self-attention layer and a cross-attention layer, which excel at capturing intra-view and inter-view relationships, respectively. The input 2D feature map is flattened from $\mathbb{R}^{C \times H_f \times W_f}$ to $\mathbb{R}^{N \times C}$, where $N = H_f \times W_f$. A position embedding, denoted as PE, is added to the flattened feature map. Fig. 5(b) illustrates the attention layer. The context refers to the input feature map itself in the self-attention layer and it represents the feature map of another view in the cross-attention layer. We use the standard multi-head attention (Vaswani et al., 2017) and layer normalization (Ba et al., 2016) in our attention layers.

## B  DATA CONFIGURATION

The synthetic images are generated by rendering objects of Objaverse from randomly sampled view-points (Liu et al., 2023). We attach these images to random backgrounds which are sampled from COCO (Lin et al., 2014). We randomly sample 128 objects from Objaverse and use 5 objects from LINEMOD sampled by Liu et al. (2022) as testing data, reserving the remaining objects for training. This design guarantees that all objects are previously unseen during the testing phase. We train the network on both synthetic and real data, alleviating the problem of domain gap.

Recall that we assume we have access to only one reference image and the objective is to estimate the relative object pose between the reference and the query. Therefore, the selection of the reference image is a crucial aspect of our benchmark. As multi-view images are available in Objaverse and LINEMOD datasets, one could randomly sample a reference given a query. However, such a strategy may yield an inappropriate reference. As shown in Fig. 6, the object depicted in the reference image barely overlaps with the one in the query, which makes the relative object pose estimation too challenging. Therefore, we filter out the inappropriate references from the datasets during training and testing, which makes our evaluation more reasonable.

Specifically, we convert the object rotation matrices $\mathbf{R}^r$ and $\mathbf{R}^q$ to Euler angles $(\alpha_r, \beta_r, \gamma_r)$ and $(\alpha_q, \beta_q, \gamma_q)$, which indicate azimuth, elevation, and in-plane rotation, respectively. Note that only azimuth and elevation lead to viewpoint changes, which thus determine the co-visible regions between the reference and query. Consequently, we set the in-plane rotation to 0 and convert the Euler angle back to the rotation matrix, i.e., $\tilde{\mathbf{R}} = h(\alpha, \beta, 0)$. We then measure the difference of the new rotation matrices $\tilde{\mathbf{R}}^r$ and $\tilde{\mathbf{R}}^q$ by computing the geodesic distance. We exclude the image pair with a distance larger than a predefined threshold ($90°$ by default in our experiments). As illustrated in Fig. 4 in our main paper, the retained image pairs display acceptable variations in object pose. Moreover, we utilize the synthetic images on Objaverse generated by Liu et al. (2023). Each 3D object model is rendered from 10 randomly sampled viewpoints, which yields synthetic images without in-plane rotations. To introduce in-plane rotations, we rotate the reference and query images using randomly sampled 2D in-plane rotations during training and testing.

Fig. 7 shows the histograms of object pose variations between the reference and query images. We measure the variations based on the geodesic distance between the two object rotation matrices $\mathbf{R}^r$ and $\mathbf{R}^q$. The histograms show that the image pairs we used in our experiments exhibit a diverse range of object pose variations, which makes our evaluation results convincing.

## C  QUALITATIVE RESULTS OF 6D OBJECT POSE ESTIMATION

We extend our method to 6D pose estimation for unseen objects by utilizing an off-the-shelf generalizable object detector (Liu et al., 2022). More concretely, instead of using dense-view reference images, we feed the one reference we have in our benchmark to the pretrained detection network, which predicts the object bounding box in the query image. We use the parameters of the object bounding box to compute 3D object translation, following the implementation in (Liu et al., 2022). Subsequently, we crop the object from the query and employ our approach to predict the relative 3D object rotation. The object rotation in the query is derived as $\mathbf{R}^q = \Delta\mathbf{R}\mathbf{R}^r$. Fig. 8 shows some qual-

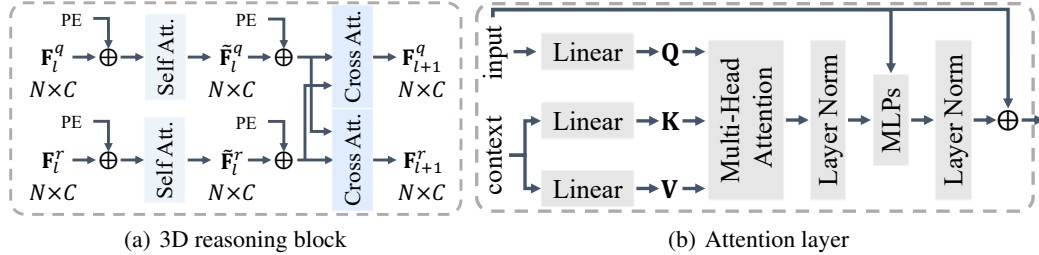

(a) 3D reasoning block        (b) Attention layer

Figure 5: **Architecture of the 3D reasoning module.**

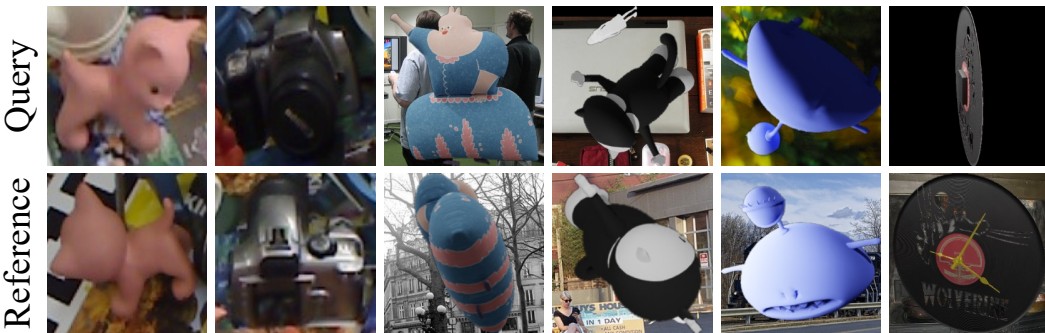

Figure 6: **Examples of inappropriate references.**

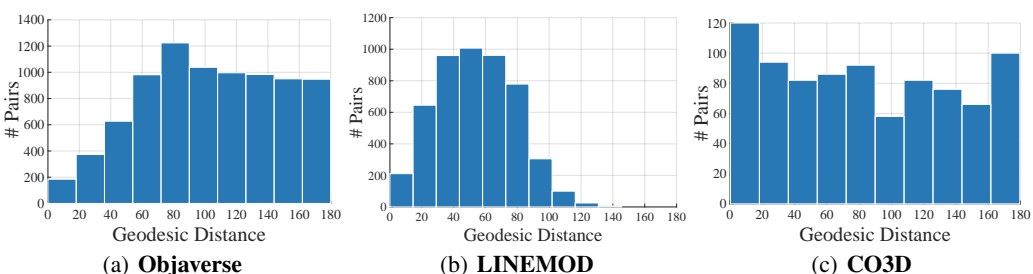

(a) **Objaverse**       (b) **LINEMOD**       (c) **CO3D**

Figure 7: **Histograms of the object pose variation between the reference and query**. We measure the object pose variation as the geodesic distance between the two object rotation matrices $\mathbf{R}^r$ and $\mathbf{R}^q$. The histogram depicts the number of image pairs falling within different distance intervals.

itative results of 6D pose estimation for the unseen objects on LINEMOD. We draw the 3D object bounding boxes in blue and green, using the predicted 6D object pose and the ground truth, respectively. The promising results demonstrate the potential of our approach in terms of generalizable 6D object pose estimation.

## D   MORE DETAIL ABOUT THE ABLATION STUDIES

As we introduced in the main paper, we performed an ablation study, evaluating the robustness against the noise added to the 2D object bounding boxes. We simulate the bounding boxes in real-world applications by performing jittering to the ground truth with different levels of noise. We denote the object center and the size of the bounding box as $c$ and $s$. We then randomly sample the perturbed parameters from the intervals $(c - 0.5 * n * s, c + 0.5 * n * s)$ and $(\frac{s}{1+n}, s * (1 + n))$, respectively, where $n$ indicates the noise. We varied $n$ from 0.05 to 0.3 in our experiments. Please refer to Fig. 5 (b) in our main paper for the experimental results.

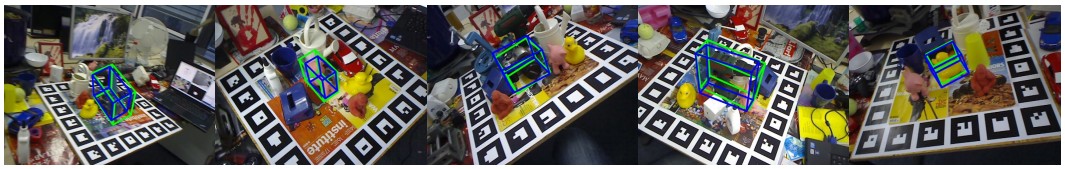

Figure 8: **Qualitative results of 6D pose estimation for unseen objects on LINEMOD.** The blue and green 3D object bounding boxes are drawn using the predicted 6D object pose and the ground truth, respectively.

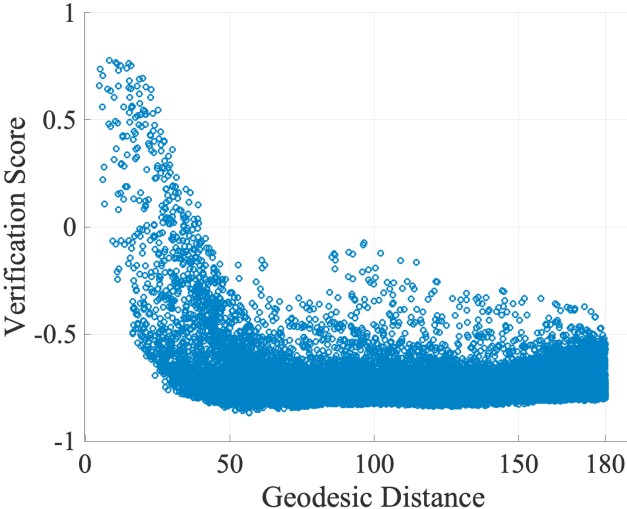

Figure 9: **Verification scores of all sampled pose hypotheses.** The x-axis and y-axis represent the geodesic distance between the pose samplings and the ground-truth relative object pose, and the verification scores, respectively.

| Method | RelPose++ | **Ours** | RelPose++-5000 | **Ours-5000** |
|---|---|---|---|---|
| MACs | 94.6 | 54.7 | 11.3 | 16.3 |
| Angular Error | 38.5 | 28.5 | 50.7 | 35.3 |

Table 5: **Efficiency.** Relpose++ uses 500,000 pose samples by default, while we sample 50,000 poses for our method in our experiments. RelPose++-5000 and Ours-5000 denote RelPose++ and our method with 5,000 pose samples, respectively. The multiply-accumulate operations (MACs) are used to measure the computation consumption.

## E  EFFICIENCY

It is worth noting that during testing, our method utilizes 50,000 pose samples, while RelPose++ uses 500,000. Despite processing fewer samples, our method achieves better accuracy in relative object pose estimation. To further evaluate the efficiency, we measure the computation cost in multiply-accumulate operations (MACs) and show the results in Table 5. All evaluated methods process the pose samples in parallel. "RelPose++-5000 and "Ours-5000" refer to RelPose++ and our method with 5,000 samples, respectively. The results clearly show that our method achieves a better trade-off between efficiency and accuracy in relative object pose estimation. Additionally, our method with only 5,000 samples still delivers more accurate results than RelPose++ with 500,000 samples.

