# OpenReview forum: "3D-Aware Hypothesis & Verification for Generalizable Relative Object Pose Estimation"
_ICLR.cc/2024/Conference — ICLR 2024 poster_

### Official Review · Reviewer_jg4A · 2023-10-30

**Soundness:** 3 good
**Presentation:** 2 fair
**Contribution:** 3 good
**Rating:** 6
**Confidence:** 3

**Summary:**

This paper proposed a 3D latent object representation and 3D-aware hypothesis verification framework for general object relative pose estimation from two 2D images. The latent 3D representation is obtained by lifting the 2D feature map into 3D space and the 3D-aware hypothesis are generated and verified in a RANSAC-like manner. Experiment has been done on public benchmark dataset CO3D, Objaverse and LINEMOD and the proposed method achieved superior performance compared with previous SOTA.

**Strengths:**

1. The proposed method combined several modules to construct an interesting pipeline for general object relative pose estimation, including 2D-to-3D feature lifting, 6D continous rotation representation hypotheses sampling, RANSAC-style hypothsis verification.
2. The achieved performance improvement over RelPos++ is impressive
3. Pose estimation robustness in case of background clutter and object motion are handled with modules of 3D masking, feature aggregation, etc.

**Weaknesses:**

1. The paper claims that the method can be used for unseen object relative pose estimation, but it actually may rely on object detection to get relevant object area for pose estimation, and there is no reliable unseen object detector available
2. The presentation is not clear and sometimes confusing, in 4.5 ablation study section, the method of 'RelPose*' should be explicitly explained, and it seems that the attention module, mask module and aggregation module donot result in significant performance improvement, it would be clearer if we start from baseline and add the proposed modules step by step to show their effectiveness
3. There are some typos, Sec.3.1, 'as illustrated in Fig.3' -> 'as illustrated in Fig.2'

**Questions:**

1. The number of hypothesis during training and testing is huge, 9000 for training and 50000 for testing, how about the computational cost? why they are different?

---

> ### Author Response · Authors · 2023-11-15
> **Response to Reviewer jg4A**
>
> We appreciate that Reviewer jg4A acknowledges our promising results and the novelty of our method. We hope the following response will address the reviewer's concerns.
>
> **Q.1 Object Detection.** In this paper, we follow the setting commonly used in previous methods [1][2][3] that focus on object rotation estimation. Nevertheless, we do consider the practical application scenario where the object bounding box is noisy. Specifically, we evaluated the robustness to this noise, and the results in Fig. 4(b) demonstrate the promising robustness of our method. Note that some of-the-shelf techniques can be used to achieve unseen object detection, such as SAM or the translation estimator presented in Gen6D. In Fig. 4 of the supplementary material, we show examples of 6D relative object pose estimates obtained by combining our method with Gen6D's translation estimator.
>
> [1] Learning descriptors for object recognition and 3d pose estimation. Paul et al., CVPR 2015.
>
> [2] Pose from Shape: Deep Pose Estimation for Arbitrary 3D Objects. Xiao et al., BMVC 2019.
>
> [3] Fusing local similarities for retrieval-based 3D orientation estimation of unseen objects. Zhao et al., ECCV 2022.
>
> **Q.2 Ablation studies.** Thanks for pointing this out. RelPose* is actually the baseline suggested by Reviewer jg4A. It aims to assess the major contribution of our paper, i.e., the 3D-aware hypothesis and verification mechanism. To this end, we maintain the backbone and the 3D representation learning module in our method unchanged, solely replacing the pose-aware verification module with the energy-based module used in RelPose and RelPose++. We denote this as RelPose*. As shown in Table 4 of the main paper, the performance of RelPose* is much worse than that of our approach, confirming that pose-aware verification is pivotal in our framework.
>
> **Q.3 Typos.** Thanks! We have corrected them.
>
> **Q.4 Number of hypotheses.** We use much fewer pose samples than RelPose and RelPose++, i.e., 50,000 for us vs. 500,000 for them. The decision to use fewer samples during training than testing was driven by the computational resource requirements for enabling mini-batch training. We evaluate the computation cost as multiply-accumulate operations (MACs) and show the results on CO3D in Table 1 of the supplementary material. Our method achieves a better trade-off between efficiency and accuracy in relative object pose estimation, delivering more accurate results with only 5,000 samples than RelPose++ with 500,000 samples.

---

> > ### Comment · Reviewer_jg4A · 2023-11-22
> >
> > Thanks the author for the detailed explanation on baseline setting and number of hypotheses comparison.  I have no further questions and will keep my positive rating

---

### Official Review · Reviewer_jWeY · 2023-10-30

**Soundness:** 3 good
**Presentation:** 3 good
**Contribution:** 3 good
**Rating:** 6
**Confidence:** 3

**Summary:**

This paper addresses the problem of relative pose estimation for unseen objects. The authors assume that only one object image as the reference is available and they aim to estimate the relative object pose between the reference and a query image. To this end, the authors propose a hypothesis-and-verification paradigm by introducing a 3D-aware verification. In particular, 3D transformation is explicitly coupled with a learnable 3D object representation. Experiments were performed on Objaverse, LINEMOD, and CO3D datasets, taking both synthetic and real data with diverse object poses into account.

**Strengths:**

Object pose estimation is a central task for 3D computer vision. Generalizable object relative pose estimation allows dealing with unseen object and is important to apply the technology to real world applications. This paper addresses this task along this direction and specifically focus on the cases that only one reference image is available. This case is difficult, especially when the object image appearance is view-dependent and the query and reference images have very different viewpoints. Despite of the difficulties, the proposed methods shows significant performance improvement than SOTA baselines on several public datasets including Objaverse, LINEMOD, and CO3D. The paper is well organized.

**Weaknesses:**

Weakness of the paper:

(1) Although the task setting chosen by the paper is challenging for SOTA methods, the motivation of this task setting is not clearly explained. Using only one reference image can pose great difficulty for the pose estimation task. An extreme example is that the reference image is taken from frontal viewpoint and a query image is taken from rear viewpoint, which may share no common features to match at all. Meanwhile, with any modern camera device, one can actually easily capture multiple images as use them as reference. In what scenarios, and why using only one reference image is critical is not mentioned in the paper.

(2) An important component of the method is the 3D reasoning model that learns the 3D volumes from 2D feature maps of two RGB image inputs. However, this is itself a very challenging research topic about open-set 3D volume representation reconstruction with two RGB inputs. It’s difficult to understand why it is possible to obtain reliable 3D representation for an unseen object from only its two views, which may even present large variations in view angle and lighting. Also, experimental verification for this component is missing. It would be informative to show some visualizations of the intermediate results of this component.

**Questions:**

Questions for the author:

(1) In what scenarios or applications is using only one reference image necessary for the pose estimation task? Why this is a meaningful problem setting?

(2) How it is possible to obtain reliable 3D volume representation for an unseen object from only its two views, which may even present large variations in view angle and lighting?

(3) Is it possible to show some visualizations of the intermediate results of the 3D reasoning component?

(4) What are the remaining error cases of the model for the defined task?

(5) What backbone network is used? Does the backbone network selection affect model performance?

(6) What is the computational cost and speed of this method?

---

> ### Author Response · Authors · 2023-11-15
> **Response to Reviewer jWeY**
>
> We thank Reviewer jWeY for the suggestions and questions. Below are our responses to each question.
>
> **Q.1 Single-reference setting.** As discussed in Sec. 2 of the main paper, existing methods for generalizable object pose estimation depend on densely sampled reference images, limiting their applicability. For instance, in scenarios such as pose estimation for non-cooperative satellites, obtaining images with diverse object poses as references is often impractical. Furthermore, in many applications, e.g., autonomous driving, only sparse viewpoints are available as references, resulting in significant object pose differences between the query and its nearest reference. In such cases, our method remains applicable since it yields promising results with a single reference, where large object pose variations are involved. Importantly, by considering the single-reference case, our work can be thought of as tackling a more general problem in the field of generalizable object pose estimation. We therefore hope that our insights will inspire future research on this important topic.
>
> Notably, predicting the relative object pose remains challenging when there is no overlap between the visible regions in the query and the reference. This issue has been thoroughly discussed in Sec. 2 and illustrated in Fig. 2 of the supplementary material. To enhance the fairness of the presented evaluation benchmark, we excluded image pairs with inappropriate references from both training and testing sets. Please refer to the supplementary material for the details.
>
> **Q.2 3D volume representation.** Unlike the mesh reconstruction task where the goal is to explicitly reconstruct a 3D object mesh from images, the 3D volume representation learned in our method serves the purpose of enabling 3D-aware pose verification. The spatial dimensionality of this representation aligns with that of the sampled rotation matrix, facilitating 3D transformations over the learned volume. Consequently, the subsequent verification is conditioned on a pose sample. In this context, pose-aware features are learned during training, making it challenging to visualize such deep features. To shed more light on the pose-aware features, we provide a detailed illustration of the verification scores for the sampled pose hypotheses in Fig. 5 of the supplementary material. In this figure, the x-axis represents the geodesic distance between the pose sample and the ground-truth relative object pose, while the y-axis indicates the verification scores. The plot reveals that the predicted score decreases as the geodesic distance becomes larger. This observation shows that our method is capable of capturing pose-aware features from the images.
>
> **Q.3 Failure cases.** As discussed in Sec. 5 of the main paper, our method may be affected by occlusions. Specifically, when computing the verification score, we estimate the local similarities of 2D feature embeddings, which preserve local geometric information. In instances of occlusion, however, some of the local information becomes unreliable. We acknowledge this as a limitation of our method and we plan to address this issue in our future work.
>
> **Q.4 Backbone.** We use MiDAS as the backbone for our method. To ensure a fair comparison with previous methods such as RelPose and RelPose++, we conducted an ablation study in Sec. 4.5 of the main paper. We replaced the introduced pose-aware verification module with the energy-based module used in RelPose and RelPose++, while keeping the backbone and the 3D representation learning module in our method unchanged. We denote this variant as RelPose* and report the results in Table 4 of the main paper. The performance considerably drops after the replacement, highlighting the critical role of the pose-aware verification module in our method rather than the backbone.
>
> **Q.5 Efficiency.** During testing, our method utilizes 50,000 pose samples, while RelPose++ uses 500,000. Despite processing fewer samples, our method achieves better accuracy in relative object pose estimation. Both methods handle the samples in parallel. The computation cost, measured in multiply-accumulate operations (MACs), is reported in Table 1 of the supplementary material. Additionally, we assess the speed of the methods on an A100 GPU. On average, our method and RelPose++ process a pair of images in 35ms and 29ms, respectively. It is worth noting that some inefficiencies, such as the use of 'for' loops, exist in our current implementation, which may impact the speed.

---

> ### Comment · Reviewer_jWeY · 2023-11-22
>
> I appreciate the authors for providing detailed responses to all my questions. I have no further questions at the moment and would like to keep my previous positive rating score.

---

### Official Review · Reviewer_c3Qb · 2023-10-30

**Soundness:** 3 good
**Presentation:** 3 good
**Contribution:** 3 good
**Rating:** 6
**Confidence:** 4

**Summary:**

The paper proposes an object pose estimator that can estimate relative pose with a single reference view.
To tackle the large viewpoint change scenarios, they propose a hypothesis-and-verification framework for robust pose estimation.
The image features and 3D feature volumes are extracted for each image first.
Then the poses are sampled and used to warp 3D feature volumes to the reference view, where the verification scores are regressed by transformer architecture. Pose is estimated by selecting the hypothesis with the largest similarity score.

**Strengths:**

1) The proposed hypothesis-and-verification framework seems to be effective especially for large viewpoint change scenarios, compared with matching-based methods.
2) The experiments and ablation studies are thorough.
3) The overall writing is clear and easy to follow.

**Weaknesses:**

1) The generalizability across datasets. The proposed method is generalizable and can handle unseen objects. I'm curious about the generalizability across datasets. For example, the proposed method is trained on the CO3D dataset, but how about the generalizability of the LINEMOD dataset? I think the generalizability across datasets is important for the proposed method, especially for real-world applications.

2) Running time comparisons. The paper needs to sample M=50000 poses for verification at test time. I'm concerned about the efficiency of estimating a single pose, even if the verifications are parallelized. I think the running time comparisons with other methods are necessary.

**Questions:**

Please refer to the weaknesses.

---

> ### Author Response · Authors · 2023-11-15
> **Response to Reviewer c3Qb**
>
> We are glad to see that Reviewer c3Qb recognizes the soundness of our approach and the promising results achieved by our method.
>
> **Q1. Generalization.** We appreciate the insightful suggestion. Since the presented benchmark includes two datasets, we did a cross-dataset experiment at the early stage of this project. This involved training a network on synthetic data from Objaverse and subsequently testing it on LINEMOD. The acc @ $30^{\circ}$ of our method and that of RelPose++ are 21.2\% and 16.7\%, respectively. Our method still achieves better performance in relative object pose estimation. However, the accuracy of both methods is lower than the results reported in our main paper. We attribute this discrepancy to factors such as the synthetic-real gap, and differences in camera intrinsic parameters and resolutions across datasets, rather than to the unseen objects. This observation aligns with discussions found in some related studies. For example, Gen6D suggests training the network on multiple datasets. In their approach, some objects from LINEMOD are chosen as training data, and all images depicting these objects are excluded during testing to ensure the scenario of unseen object pose estimation. To shed more light on this finding, we mimicked the setting in Gen6D by fine-tuning our method for only 200 steps on the same subset of LINEMOD using the model pretrained on Objaverse. This took only around two minutes. Note that, in this case, the objects from LINEMOD during testing are still unseen. The acc @ $30^{\circ}$ significantly increases from 21.2\% to 60.1\%. While these results suggest that fine-tuning effectively mitigates the domain gap, we agree with the reviewer that cross-domain generalization is an important and interesting research problem. We nonetheless leave it as future work.
>
> **Q.2 Running time.** Thanks for the suggestion! Compared to RelPose++ which requires 500,000 pose samples, our method only uses 50,000 while achieving much better accuracy in relative object pose estimation. In response to the reviewer's suggestion, we assessed the runtime performance of our method and RelPose++ on an A100 GPU. Both of these two methods process the pose samples in a parallel manner. On average, our method and RelPose++ take 35ms and 29ms, respectively, to process a pair of images. Due to some inefficiencies in our current implementation such as 'for' loops, our method is slightly slower than RelPose++. However, the sacrifice in running time is deemed acceptable given the superior accuracy achieved by our method.

---

> > ### Comment · Reviewer_c3Qb · 2023-11-22
> >
> > Thanks for the clarifications. I have no extra questions and I keep my original positive rating.

---

### Official Review · Reviewer_nmWB · 2023-10-31

**Soundness:** 4 excellent
**Presentation:** 4 excellent
**Contribution:** 2 fair
**Rating:** 6
**Confidence:** 4

**Summary:**

The paper addresses the problem of estimating the pose of an object observed in a query image relative to a reference image. A novel method is presented for estimating the relative 3D orientation of the object. Multiple orientation hypotheses are sampled and scored using the proposed verification strategy. The verification strategy relies on extracting 3D feature volumes from the query and reference image, and measuring similarity between the 3D volume of the query with the 3D volume of the reference transformed with the 3D orientation of the hypothesis. The comparison of the 3D volumes is done using a novel feature aggregation strategy to gain robustness with respect to the background. The paper also introduces a new benchmark for this task, relying on synthetic images of objaverse objects, and real images of the LINEMOD dataset.

**Strengths:**

S1. The paper is well easy written and easy to understand.

S2. The method can operate from RGB images, and only requires a single reference image.

S3. The technical contributions are sound and demonstrated to be effective in an ablation study (Table 4 of the main paper). In particular, the attention layers to extract 3D features capturing information from both query and reference images; and the aggregation mechanism to gain robustness with respect to the background are well motivated.

S4. The method combines several interesting design choices which are demonstrated to lead to higher relative rotation estimation accuracy compared to several baselines (Table 1). However it is not clear if the comparison with RelPose and RelPose++ is fair (see weakness W3).

**Weaknesses:**

W1. The object pose estimation setup addressed in this paper should be better motivated. If a CAD model of the object is available, multiple reference images can be rendered or other methods applied directly, e.g. [A,B,C,D,E,F]. In this paper, availability of a CAD model of the object is not assumed, but obtaining the pose of the object with respect to the camera in the query image would still require the pose of the object to be known in the reference image. It is not clear how the reference pose would be obtained in practice without assuming the CAD model is known. In addition, the method presented and evaluated in this paper does not estimates the absolute or relative 3D translation of the object between the query and reference images. Overall, the paper would be more convincing if practical applications of the presented approach were discussed.

[A] DeepIM: Deep Iterative Matching for 6D Pose Estimation. Li et al, ECCV 2018.

[B] Pose from Shape: Deep Pose Estimation for Arbitrary 3D Objects. Xiao et al, 3DV 2019.

[C] Multi-path Learning for Object Pose Estimation Across Domains. Sundermeyer et al, CVPR 2020.

[D] CorNet: Generic 3D Corners for 6D Pose Estimation of New Objects without Retraining. Pitteri et al, ICCW 2021.

[E] OSOP: A Multi-Stage One Shot Object Pose Estimation Framework. Shugurov et al, CVPR 2022.

[F] MegaPose: 6D Pose Estimation of Novel Objects via Render & Compare. Labbe et al, CoRL 2022.

W2. The runtime of the approach is not mentionned. 50000 hypotheses are evaluated during testing, what is the tradeoff between accuracy and runtime ?

W3. Is it not clear if the RelPose, RelPose++ and the presented approach were all trained using the same images for the comparison on CO3D presented in Table 1. The supplementary material mentions that the approach is trained using objects from the objaverse dataset. Is it also the case for the experiments in Table 1 ?

Minor details:
- Some papers have incorrect citations, e.g. Gen6D is an ECCV paper, DOPE is a CoRL paper.

**Questions:**

The method presented in this paper is well designed and demonstrated to be effective for estimating the relative 3D orientation of an object observed in two different images. I however currently have questions regarding the quantitative comparison with state-of-the-art (W3), and concerns regarding the practical applications of such approach (W1) and it’s runtime (W2). I will gladly increase my rating if these concerns can be addressed.

---

> ### Author Response · Authors · 2023-11-15
> **Response to Reviewer nmWB**
>
> We sincerely thank Reviewer nmWB for acknowledging our technical contributions. Below are detailed responses to the reviewer's concerns:
>
> **Q1. Single-reference setting.** It is indeed often impractical to assume the availability of the textured 3D mesh model for a novel object, and we therefore acknowledge the contributions of the model-free approaches, including the ones listed by the reviewer. The missing references have been added to the main paper. To reasonably define the pose of an unseen object, these methods rely on multi-view calibrated reference images. Let us take Gen6D as an example since methods such as Pose from Shape [B] and OSOP [E] still require the 3D object model to generate the references. Gen6D exploits real reference images that are calibrated using a structure-from-motion technique. The goal then is to estimate the relative object pose between the query and the most similar reference selected from all multi-view reference images. Our method addresses a more general and challenging scenario where only a single reference is available. In this case, the potentially large object pose difference between the query and the reference makes methods such as Gen6D inapplicable, whereas our method can still predict the relative object pose. Moreover, in our experiments, we show that, when the reference object pose is known, such as in the CO3D, Objaverse, and LINEMOD datasets, we can exploit this information to derive the query object pose. Nevertheless, our method can still be applied to scenarios where the reference is not calibrated. For example, RelPose and RelPose++ introduce a scenario of sparse-view 3D reconstruction for unseen objects, where our method can serve as a stepping stone. In this context, one view act as reference and its rotation is set to the identity matrix. The pairwise relative transformations obtained by our approach can be leveraged to reconstruct the object's 3D structure.
>
> **Q2. Translation estimation.** Rotation estimation is acknowledged to be more challenging than translation estimation because it involves computing an element $\mathbf{R}\in SO(3)$, and as such, it has received more attention [1][2][3].
> As discussed in Gen6D, object translation estimation is facilitated by 2D object detection, and zero-shot object detection can be achieved using Gen6D's detector or other techniques such as SAM. In our paper, we conducted an experiment to evaluate the robustness of our approach to the object bounding box. As shown in Fig. 4 (b), our method achieves better robustness than other evaluated approaches, thus showing its potential to be integrated with such object detectors. In Fig. 4 of the supplementary material, we illustrated some 6D object pose estimation results, combining our method and the translation estimator introduced in Gen6D.
>
> [1] Learning descriptors for object recognition and 3d pose estimation. Paul et al., CVPR 2015.
>
> [2] Pose from Shape: Deep Pose Estimation for Arbitrary 3D Objects. Xiao et al., BMVC 2019.
>
> [3] Fusing local similarities for retrieval-based 3D orientation estimation of unseen objects. Zhao et al., ECCV 2022.
>
> **Q.3 Efficiency.** Our method requires much fewer pose hypotheses than the state-of-the-art RelPose++ while achieving more accurate results. Specifically, RelPose++ samples 500,000 poses during testing while we only use 50,000. For both RelPose++ and our method, we compute the verification scores for all pose samples in a parallel manner during testing, so we measure the computation cost as multiply-accumulate operations (MACs) and report the results on CO3D in Table 1 of the supplementary material. The results clearly show that our method achieves a better trade-off between efficiency and accuracy in relative object pose estimation. Our method with only 5,000 samples still delivers more accurate results than RelPose++ with 500,000 samples. The average processing time on an A100 GPU for our method and RelPose++ is 35ms and 29ms, respectively. It is important to note that our current implementation contains certain inefficiencies, including the use of 'for' loops, which may influence the overall speed.
>
> **Q.4 Comparison on CO3D.** All evaluated methods were compared fairly. We strictly followed the setting in RelPose++, training and testing our method on the same data as RelPose and RelPose++. The results reported in Table 1 of our main paper are comparable to the ones shown in Table 1 of RelPose++. On Objaverse and LINEMOD, we also conducted a fair comparison, using the same training and testing data for all methods.
>
> **Q.5 Minor details.** Thanks for pointing this out. We have updated our paper accordingly.

---

> > ### Comment · Reviewer_nmWB · 2023-11-20
> >
> > Thank you for the detailed answer.
> >
> > Q1. I acknowledge that predicting the relative pose using a single reference image is a challenging problem. In CO3D, Objaverse and LINEMOD datasets, the reference frame is annotated with the object pose, but my concern is whether obtaining this reference pose can be obtained in a similar setting where only a single image and no CAD model are available. The applications of RelPose/RelPose++ to sparse-view 3D reconstruction of unseen objects is a more convincing example to motivate the setting. I encourage the authors to mention it in the paper in order to better convince the reader of the practical relevance of the single-reference/no cad model setting.
> >
> > Q2. 3D translation can be estimated from a 2D bounding box, but the result is sensitive to the 2D detection error. Thank you for pointing to Fig 4 of the supplementary material where the 3D translation is estimated. The second image (from the left) shows a noticeable translation error. I would be more convinced if the 3D translation error of the proposed approach was compared with alternatives, e.g. RelPose++.
> >
> > Q3, Q4. Thank you, my concerns have been addressed.
> >
> > My concerns regarding the runtime of the approach and the comparison on CO3D have been addressed. Given the significant improvements over RelPose++ (in terms of rotation accuracy) and the comparable runtime, I have increased my rating.

---

### Meta-Review · Area_Chair_6b38 · 2023-12-08

**Metareview:**

This paper tackles the task of relative rotation estimation given two object-centric images, and proposes an energy-based approach for this task. Unlike prior work (RelPose, RelPose++) which adopts a similar overall framework, this paper introduces a 3D reasoning module for computing the score. Specifically, it computes per-image 3D features that are rotated based on a relative pose hypothesis and then used to infer a score via dot product. The empirical results and extensive ablations clearly show the benefit of this design choice.

This paper received all borderline positive reviews. In particular, the reviewers all appreciated the empirical results (across three datasets) as well as the principled design of the energy-based model. However, there are still some concerns regarding the approach e.g. inability to estimate translation (it relies on an off-the-shelf system), demonstrations restricted to 2-view input (as opposed to the general n-view settings in prior work). Moreover, the technical contribution is also perhaps limited as the overall framework is adapted from prior work (with a new score prediction module).

Overall, the AC was slightly torn about the work, but felt that the principled design of the approach and the strong empirical results across multiple datasets outweighed the concerns above and would lean towards acceptance.

**Justification For Why Not Higher Score:**

There are still some concerns regarding the approach e.g. inability to estimate translation (it relies on an off-the-shelf system), demonstrations restricted to 2-view input (as opposed to the general n-view settings in prior work). Moreover, the technical contribution is also perhaps limited as the overall framework is adapted from prior work (with a new score prediction module).

**Justification For Why Not Lower Score:**

Overall, the principled design of the approach and the strong empirical results across multiple datasets outweighed the concerns above.

---

### Decision · Program_Chairs · 2024-01-16

Accept (poster)